# Non-equilibrium steady states in supramolecular polymerization

Alessandro Sorrenti[1], Jorge Leira-Iglesias[1], Akihiro Sato[1] & Thomas M. Hermans[1]

Living systems use fuel-driven supramolecular polymers such as actin to control important cell functions. Fuel molecules like ATP are used to control when and where such polymers should assemble and disassemble. The cell supplies fresh ATP to the cytosol and removes waste products to sustain steady states. Artificial fuel-driven polymers have been developed recently, but keeping them in sustained non-equilibrium steady states (NESS) has proven challenging. Here we show a supramolecular polymer that can be kept in NESS, inside a membrane reactor where ATP is added and waste removed continuously. Assembly and disassembly of our polymer is regulated by phosphorylation and dephosphorylation, respectively. Waste products lead to inhibition, causing the reaction cycle to stop. Inside the membrane reactor, however, waste can be removed leading to long-lived NESS conditions. We anticipate that our approach to obtain NESS can be applied to other stimuli-responsive materials to achieve more life-like behaviour.

[1] University of Strasbourg, CNRS, ISIS UMR 7006, F-67000 Strasbourg, France. Correspondence and requests for materials should be addressed to T.M.H. (email: hermans@unistra.fr).

Dissipative supramolecular polymerization is commonplace in living cells such as in the actin and microtubule networks, where chemical fuels control spatiotemporal behaviour, endowing the cell with exquisite responsivity, motility, adaptability and replication[1]. In the cell, mitochondria continuously deliver the chemical fuels such as ATP to the cytosol, where the energy-consuming processes take place, while removing waste products like ADP and Pi (phosphate ions)[1].

Artificial supramolecular assemblies have been developed in the recent years where a monomer can be switched from assembling to nonassembling or vice versa by direct chemical modification[2–4], by enzymes[5–8] or mediated by chelation of nucleobases[9–12]. More recently, this has resulted in transient assemblies for which the lifetime of the assembled state can be programmed in advance. Moreover, these systems can be reactivated by addition of a fresh aliquot of the chemical fuel, resulting in a new cycle of transient assembly[2,3,6,11–13] (also called a batch excursion in the phase space[14]). However, waste from the fuel conversion is not usually removed from the system that over longer times interferes with the assembly process. The latter results in poisoning of the system, and a reduced response to addition of more fuel.

Here we demonstrate that it is possible to keep supramolecular polymers in various sustained non-equilibrium steady states (NESS) by a continuous influx of chemical fuel and outflux of waste products. We control our supramolecular polymerization by phosphorylation/dephosphorylation mediated by kinase and phosphatase enzymes, respectively. The phosphorylation is driven by the fuel ATP that adds a charged phosphate group to our self-assembling molecule resulting in polymer growth and switching of supramolecular chirality. We show that in batch conditions transient self-assembly occurs upon addition of ATP, but poisoning due to phosphatase inhibition limits the number of assembly/disassembly cycles. The poisoning can be overcome by using a home-built membrane reactor, where different NESS can be maintained depending on the level of fuel supplied and waste removed under continuous flow conditions. The result is that the assembly/disassembly process can be kept continuously working for days as long as the fuel is supplied. Maintaining NESS conditions has been heralded as one of the requirements to obtain life-like systems and materials[15].

## Results

**Designing a switchable supramolecular polymer.** A symmetric peptide derivative of 3,4,9,10-perylenediimide (**PDI** in Fig. 1a), containing the consensus sequence LRRASL for protein kinase A (PKA)[16], was prepared as reported in the Supplementary Methods and Supplementary Figs 1–3. **PDI** can be phosphorylated at both serine residues by PKA consuming ATP, resulting first in **p-PDI** and finally in **p2-PDI** (Fig. 1a). The two phosphate groups of **p2-PDI** can be covalently cleaved (Fig. 1a, scissors) by λ-protein phosphatase (λPP), first to **p-PDI** and then to **PDI**. Two equivalents of ADP and inorganic phosphate (Pi) are produced as waste products in a complete phosphorylation/dephosphorylation cycle (Fig. 1a). Phosphorylation of serine (or threonine) is used in living systems to control chromatin structures and signal transduction, and has been previously implemented in the design of artificial (equilibrium) switchable polymeric assemblies[17] and supramolecular hydrogelators[5,7]. A schematic overview of the enzyme-controlled supramolecular system is reported in Supplementary Fig. 4, where the complete enzyme network is depicted.

Perylenediimide derivatives are prone to self-assemble in water due to $\pi - \pi$ stacking, the hydrophobic effect and electrostatic interactions[18,19]. The introduction of two phosphate groups onto

our molecule changes its charge ($4^+$ for **PDI** versus nearly zwitterionic $4^+/3.3^-$ for **p2-PDI**), in turn affecting the structure and stability of their supramolecular polymers.

Our approach, summarized in Fig. 1a, guarantees that the activation/deactivation of the self-assembling molecules proceed via different chemical pathways (that is, mediated by two different enzymes working at the same time) that has been argued to be a requirement to obtain NESS[20]. The conversion between the three species (**PDI**, **p-PDI** and **p2-PDI**) occurs through irreversible enzyme-catalysed reactions. Thus, the solution composition is only dictated by the relative rates of competing phosphorylation and dephosphorylation, but not by their thermodynamic stability (that is, the three species are not in chemical equilibrium, cf., red crosses in Fig. 1a). Moreover, to have sustained NESS it is crucial to have fast and completely reversible cycles of (de)activation. Otherwise, the reaction cycle **PDI→p-PDI→p2-PDI→p-PDI→PDI** would stop after several runs. We therefore introduced triethylene glycol in the **PDI** structure to increase water solubility and decrease bundling and/or gelation that has previously hampered accessibility of enzymes to the target sites[5,7]. Our studies first show that we can obtain a stepwise response (Fig. 1b) in analogy with known stimuli-responsive materials[21]. Next, we can obtain transient states (Fig. 1c), and finally sustained NESS (Fig. 1d) using a membrane reactor, as will be described below.

We first studied the phosphorylation of **PDI** by liquid chromatography–mass spectrometry (LC–MS) and $^{31}$P-nuclear magnetic resonance to verify that PKA can indeed quantitatively phosphorylate **PDI** to **p2-PDI**, using two equivalents of ATP (Supplementary Figs 5 and 6). In fact, phosphorylation might not work, or be too slow on a reasonable timescale, in synthetic molecules bearing only short peptide (consensus) sequences[22]. In a second step, the two phosphate groups of **p2-PDI** can be quantitatively cleaved upon addition of λPP yielding again nonphosphorylated **PDI** (Supplementary Fig. 7). Careful choice of the available phosphatases had to be done to prevent inhibition due to the triethylene glycol tails (see Supplementary Discussion/Supplementary Fig. 8).

Beyond verifying the ability of PKA and λPP to catalyse the (de)phosphorylation of **PDI**, it is nontrivial to find experimental conditions (buffer composition, pH, enzyme and substrate concentrations) at which both the enzymes can work, and at commensurate speeds. The optimized buffer is a compromise between the optimal ones for each of the two enzymes and is referred to as the reaction buffer (see Methods).

**Effect of phosphorylation on supramolecular polymerization.** We investigated the self-assembly behaviour of **PDI** and **p2-PDI** in the reaction buffer to determine the changes induced by phosphorylation on their supramolecular polymers. To this end, we performed phosphorylation and dephosphorylation in two separate steps (Fig. 1b). **PDI** dissolves readily in buffer, even in the mM range, giving homogeneous cherry-red solutions stable over time. The ultraviolet–visible spectrum of 200 μM **PDI** solutions displays a broad absorption band (400–620 nm) due to the S0 → S1 transition of the perylenediimide chromophore, with two maxima at 504 and 540 nm ($A_{504\,nm}$ and $A_{540\,nm}$) (Fig. 2a). The ratio $A_{504\,nm}/A_{540\,nm}$ of 1.46 strongly indicates **PDI** aggregation through π–π stacking (H-aggregation) into helical stacks of molecules with rotational offset between them (see below)[18,19,23]. As a reference, monomeric **PDI** has a $A_{504\,nm}/A_{540\,nm}$ ratio of 0.65 (Supplementary Fig. 9). Upon addition of PKA ($c = 0.13\,μM$) and ATP ($c = 400\,μM$, 2 equivalents (eq.)) to the **PDI** solution (200 μM), a continuous increase of $A_{504\,nm}/A_{540\,nm}$ over time was observed, up to a

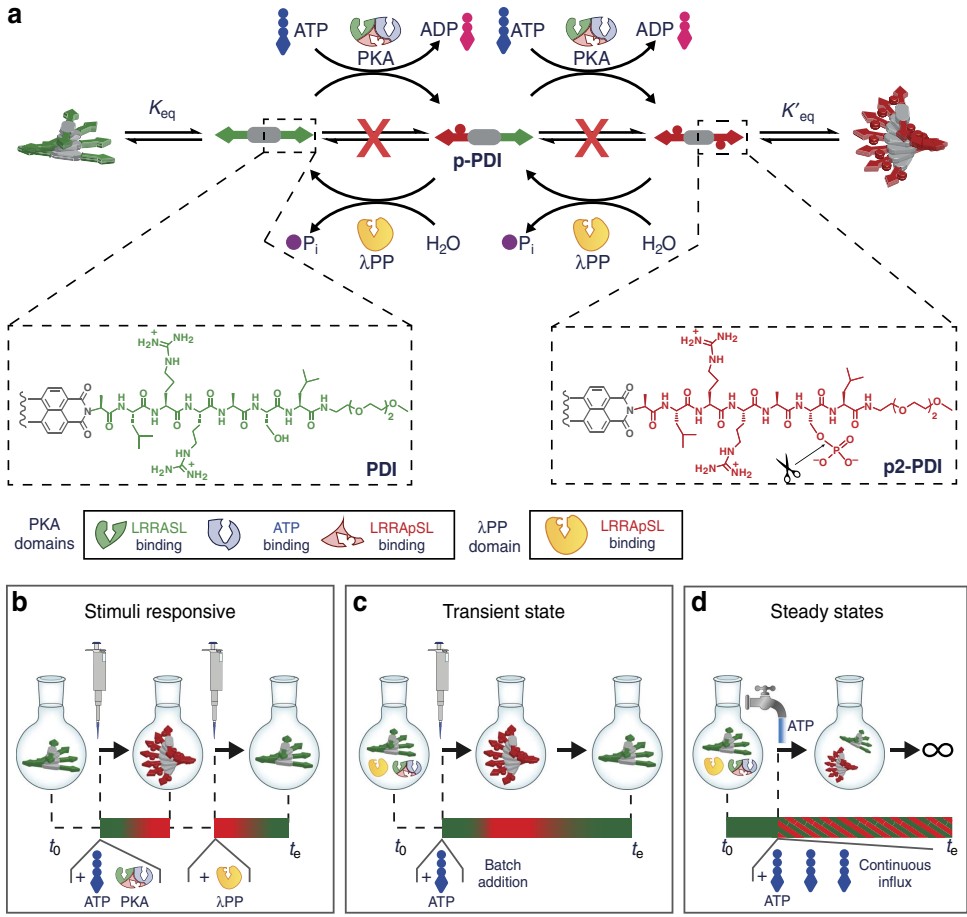

**Figure 1 | Enzyme-controlled supramolecular polymerization from stepwise to steady states.** (**a**) Peptide–perylenediimide derivative **PDI** (half is shown) is phosphorylated on the serine residue by protein kinase A (PKA) to give monophosphorylated **p-PDI**, and further diphosphorylated **p2-PDI**, fuelled by ATP to ADP hydrolysis (one eq. per phosphate introduced). Phosphate hydrolysis (scissors) by λ-protein phosphatase (λPP) yields inorganic phosphate Pi as waste. Both **PDI** and **p2-PDI** can self-assemble to form equilibrium supramolecular polymers. PKA has three binding sites: for ATP (blue), for the LRRASL peptide (green) and for LRRApSL (red). (**b**) Stimuli responsiveness: the addition of ATP and PKA to a solution of **PDI** results in **p2-PDI** and a consequent change of the supramolecular structure of the polymer. A second stimulus, that is, the addition of λPP, is needed to reset the polymer to its original nonphosphorylated state. (**c**) Transient state: a single input, that is, the addition of ATP to a solution of **PDI**, in the presence of PKA and λPP, leads to a transient change of the supramolecular structure and chirality of the polymer. (**d**) Supramolecular non-equilibrium steady states (NESS). The system is kept in a dissipative steady state by continuous influx of ATP. Depending on the level of the chemical fuel supplied different dissipative steady states can be accessed.

plateau at 1.65 after ∼100 min (Fig. 2b). LC–MS analysis of the solution at the plateau confirmed the quantitative conversion of **PDI** to **p2-PDI**. The observed increase of $A_{504\,nm}/A_{540\,nm}$, and the overall decrease in intensity (Fig. 2a) indicates that phosphorylation promotes further aggregation[18,24] as well as growth of the aggregates into bigger structures (see below). A negative control where only ATP was added did not result in any change with time of $A_{504\,nm}/A_{540\,nm}$.

The mechanism of supramolecular polymerization for both **PDI** and **p2-PDI** was studied by temperature-dependent ultraviolet–visible experiments (see Supplementary Discussion). The latter showed in both cases a continuous decrease of the ratio $A_{504\,nm}/A_{540\,nm}$ upon heating from 283 to 368 K, indicative of disassembly (Supplementary Fig. 10a). The observed spectral changes were reversible upon consecutive heating–cooling cycles (at a rate of $1\,K\,min^{-1}$), suggesting that the polymers are equilibrium structures with exchange dynamics on the second timescale. The $A_{504\,nm}/A_{540\,nm}$ versus temperature curves could be described with an isodesmic (equal-K) polymerization model[25,26] for both **PDI** and **p2-PDI**. This analysis (Supplementary Discussion/Supplementary Table 1) shows that **p2-PDI** polymers

($K_{eq}$ (298 K) = $3.8 \times 10^4\,M^{-1}$, $T_m$ (230 μM) = 368.9 ± 0.1 K) are more stable as compared with **PDI** polymers ($K_{eq}$ (298 K) = $2.1 \times 10^4\,M^{-1}$, $T_m$ (230 μM) = 358.5 ± 0.1 K). Perylenediimide derivatives are well known for their propensity to assemble into one-dimensional columnar aggregates[18,19,27]. In the case of **p2-PDI** we could observe bundles of long fibrous aggregates by atomic force microscopy (μm-sized, 1 ± 0.2 nm height for the smallest fibres) in samples drop cast on mica (Fig. 2d and Supplementary Fig. 11). Shorter fibres could also be imaged by transmission electron microscopy (Supplementary Fig. 12). However, our main focus was on assessing the changes induced by phosphorylation on the supramolecular polymer in the reaction buffer. To this end, we performed dynamic light-scattering (DLS) measurements that confirmed the increase in the polymer size upon phosphorylation. As shown in Fig. 2c, the normalized cross-correlation function $g^{(1)}(\tau)$ of **p2-PDI** (red squares) is clearly shifted to higher lag time $\tau$ when compared with that of **PDI** (black circles), corresponding to hydrodynamic radii $R_H$ of 690 ± 40 and 440 ± 10 nm for **p2-PDI** and **PDI**, respectively (at 1 mM). In addition, the corresponding relaxation Γ showed a slope of 2 versus scattering vector squared

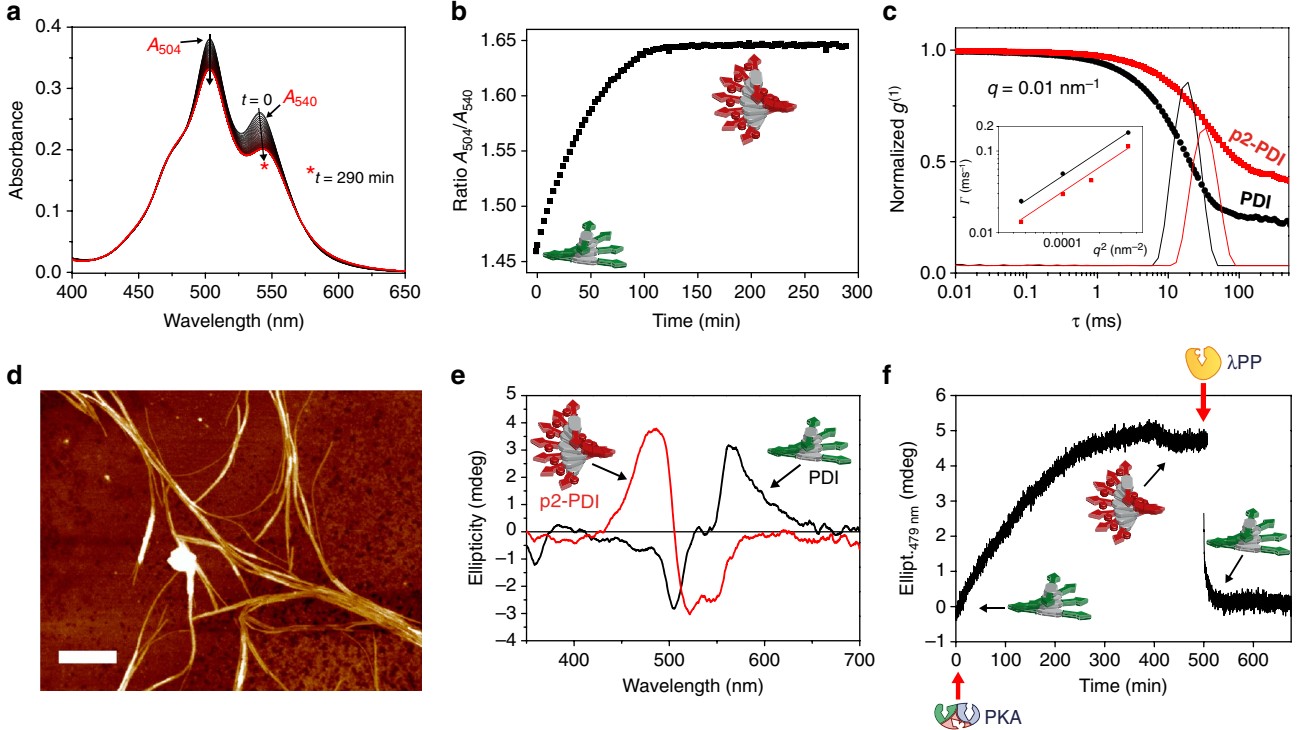

**Figure 2 | Changes in the supramolecular polymer induced by (stepwise) phosphorylation. (a)** Ultraviolet–visible spectra of a 200 μM **PDI** solution (optical path 1 mm) during phosphorylation triggered by the addition of PKA (0.13 μM) and ATP (400 μM), showing the evolution of the two main peaks ($A_{504\,nm}$, $A_{540\,nm}$) (black arrows). **(b)** Time-course $A_{504\,nm}/A_{540\,nm}$ during phosphorylation, extracted from spectra in **a**. **(c)** DLS measurements: normalized crosscorrelation function ($g^{(1)}(\tau)$) of 1 mM **PDI** (black dotted line) and **p2-PDI** (red dotted line) solutions, showing a shift to higher lag-time $\tau$ upon phosphorylation. The corresponding distribution of the relaxation times for **PDI** (black solid line) and **p2-PDI** (red solid line). The insert: relaxation $\Gamma$ versus the square of the scattering vector $q^2$. **(d)** Atomic force microscopy (AFM) image of 2 μM **p2-PDI** drop cast on mica. Scale bar, 500 nm. **(e)** CD spectra of 200 μM **PDI** (black line) and **p2-PDI** (red line) solutions (optical path 1 mm). **(f)** Time-course CD measurements (ellipticity at 479 nm versus time) of a 200 μM **PDI** solution during stepwise phosphorylation/dephosphorylation (optical path 1 mm, 400 μM ATP, 0.13 μM PKA, followed by 0.3 μM λPP at $t = 500$ min).

$q^2$ (log-log insert in Fig. 2c), confirming that these processes are diffusive.

We explain the observed increase in size and stability of the supramolecular assemblies upon phosphorylation by considering the overall charge of the monomer. Specifically, **p2-PDI** is nearly polyzwitter-ionic $4^+/3.3^-$ (as calculated from the $pK_a$ values) that is known to lower solubility and diminish electrostatic repulsion, leading to increased supramolecular polymerization.

When studied by circular dichroism (CD) spectroscopy, we found that **PDI** solutions feature an asymmetric positive CD couplet at the main absorption of the perylenediimide chromophore (black line in Fig. 2e) that is due to right-handed P-helical stacking[28–31]. Interestingly, an inversion of the CD spectrum to a clear negative couplet was observed upon phosphorylation, pointing to left-handed M-helical polymers for **p2-PDI** (red line in Fig. 2e)[28–31]. In addition, coexistence of separate **PDI** and **p2-PDI** polymers in solution (important later on) was verified by self-sorting experiments[32,33]. Specifically, CD spectra were found to be linear combinations of the individual **PDI** and **p2-PDI** spectra corresponding to their molar fraction, suggesting that they do not form mixed assemblies[32,33] (Supplementary Fig. 13). In other words, phosphorylation leads to self-sorted **p2-PDI** polymers that not only have a higher degree of polymerization, but also opposite helicity (also observed recently in naphthalene derivatives[12]).

**Stepwise phosphorylation and dephosphorylation.** Due to the unique CD spectra of **PDI** and **p2-PDI** we could use

time-dependent CD measurements to follow the phosphorylation and dephosphorylation in a stepwise manner (Fig. 1b). For instance, the addition of PKA ($c = 0.13$ μM) to a solution of **PDI** ($c = 200$ μM) and ATP ($c = 400$ μM, 2 eq.) resulted in a progressive change (inversion) of the CD spectra over time (Supplementary Fig. 14). The latter resulted in an increase of ellipticity at 479 nm until a plateau was reached at reaction completion (Fig. 2f). After complete phosphorylation (checked by LC–MS), the addition of λPP ($c = 0.3$ μM, at $t = 500$ min) caused a fast decrease of the ellipticity at 479 nm to its initial value, with the CD spectrum reverting to the shape characteristic for **PDI** (again LC–MS confirmed complete dephosphorylation). Note that the latter occurred in the absence of ATP, since it was completely consumed in the prior phosphorylation. So far, we have shown that it is possible to reversibly phosphorylate and dephosphorylate our self-assembling peptide-perylenediimide derivatives, and that phosphorylation leads to longer polymers with opposite chirality that coexist in solution with nonphosphorylated polymers.

**Transient states in the supramolecular structure of the polymer.** Encouraged by our stepwise phosphorylation/dephosphorylation experiments, we optimized the reaction buffer to achieve a full cycle (that is, **PDI** → **p-PDI** → **p2-PDI** → **p-PDI** → **PDI**) in a nonstirred batch reactor (cf. Fig. 1c). Figure 3a shows an exemplary transient state in the fraction of **p2-PDI**, that is, a rapid increase followed by a slower decrease upon addition of a shot of ATP ($c = 1$ mM, 10 eq.) to a **PDI** solution ($c = 100$ μM)

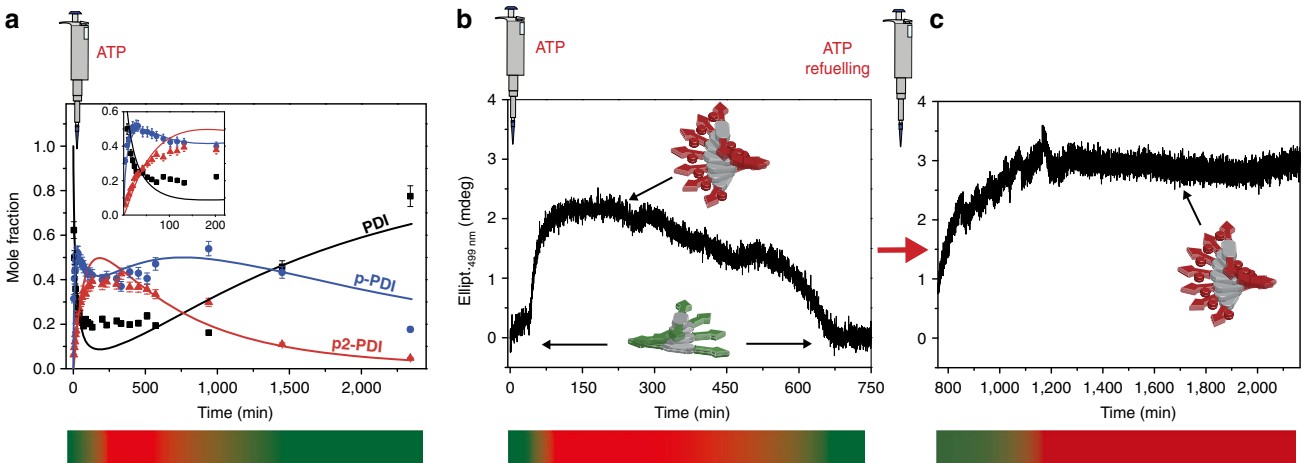

**Figure 3 | Transient state and refuelling.** (**a**) Molar fraction of **PDI**, **p-PDI** and **p2-PDI** versus time by LC–MS (points), following the batch addition of ATP (1 mM), PKA (0.13 μM) and λPP (0.3 μM) to a 100 μM **PDI** solution. The solid lines are the fits of the ODE model. The insert is a zoom at the short times. (**b**) Time-course CD measurements (monitored at 499 nm, optical path 1 mm) following the batch addition of ATP (2 mM) to a 200 μM **PDI** solution containing PKA (0.13 μM) and λPP (0.2 μM). The coloured bars at the bottom of **a,b** indicate the transient formation/disappearance of **p2-PDI** (red colour) starting from **PDI** (green colour). (**c**) Time-course CD measured after refuelling the same solution as in **b** with a second shot of ATP (2 mM). In this case **p2-PDI** accumulates.

containing PKA ($c = 0.13$ μM) and λPP ($c = 0.3$ μM). Interestingly, a rather complex behaviour was observed for monophosphorylated **p-PDI**, showing two maxima during a single transient experiment (Fig. 3b, blue line). To rationalize the latter behaviour, we developed a mathematical model of the enzymatic network depicted in Supplementary Fig. 4, using ordinary differential equations to describe the mass-action dynamics (Supplementary Discussion/Supplementary Tables 2 and 3). Our model takes into account the detailed kinetic mechanism of phosphorylation by PKA and dephosphorylation by λPP, as well as the possible (reversible) inhibition reactions that can occur at the expense of the two enzymes. In addition, the presence of two phosphorylation sites for each **PDI** molecule is considered, resulting in a network of 55 chemical species and an equal number of ordinary differential equations, Supplementary Fig. 15. Many of the PKA rates have been previously reported in the literature[34,35]. However, the conditions used in this work deviate from the latter, since we optimized the reaction buffer to have comparable rates for PKA and λPP. To fit the reaction rates to our conditions we performed a global minimization procedure (Supplementary Discussion/Supplementary Table 4) that resulted in good agreement of **PDI**, **p-PDI** and **p2-PDI** concentrations (solid lines, Fig. 3a) with the experimental results (points, Fig. 3a). A closer look at the reaction rates calculated by the model showed that the first maximum of **p-PDI** (Fig. 3a, $t = 37$ min) is due to the competition between the two phosphorylation reactions **PDI→p-PDI** and **p-PDI →p2-PDI**, while the second maximum ($t \approx 750$ min) results from the competition between the phosphorylation and dephosphorylation of **p2-PDI**, that is, **p-PDI→p2-PDI** and **p2-PDI→p-PDI** (Supplementary Fig. 16). For a similar transient experiment, CD measurements showed a transient increase of ellipticity at 499 nm that then slowly decayed to its original value (Fig. 3b). This clearly demonstrates that a transient change of the supramolecular structure and chirality of the noncovalent polymer can be obtained that is energetically driven by the consumption of ATP. In other words, the perylenediimide assemblies are pushed transiently out of equilibrium. The time evolution is controlled by the rates of competitive enzymatic phosphorylation/dephosphorylation dictated by the complex dynamics of the network. Once all the fuel is consumed, phosphorylation is no longer possible, and the

system relaxes back to its equilibrium nonphosphorylated state, that is, to **PDI**. The model also allows to explore a more extended phase space as a function of the concentrations of PKA, λPP and ATP (Supplementary Fig. 17).

In analogy with recent studies[2,6,11,12] we explored the possibility to perform multiple transient experiments by refuelling with ATP. However, when adding a second aliquot of ATP to the same solution (Fig. 3b, see description above), we could only observe phosphorylation but not dephosphorylation back to **PDI** (Fig. 3c). Looking at the enzymatic network (Supplementary Fig. 4) we can see that a full cycle (that is, **PDI→p-PDI→p2-PDI→p-PDI→PDI**) consumes two equivalents of ATP, and produces the same molar amount of ADP and phosphate Pi. The latter is a potent λPP inhibitor[36] that likely prevents the dephosphorylation of **p2-PDI**. The poisoning due to accumulation of waste is a ubiquitous problem for chemically fuelled supramolecular systems in (semi)batch conditions[2,6,11,12]. To resolve the poisoning problem we dialysed the solution (Fig. 3c, $t = 2,160$ min) against fresh buffer, choosing a membrane with a suitable molecular weight cutoff (MWCO = 2 kD). Encouragingly, it was possible to get rid of the waste (ADP + Pi), and as a result the system returned back completely to the starting nonphosphorylate state **PDI** as confirmed by LC–MS and CD (Supplementary Fig. 18). In addition, the latter dialysis experiments also proved that cross-deactivation between the two enzymes does not take place, and the network keeps working perfectly for days.

**Supramolecular NESS**. Adding fuel and removing waste is customary when studying chemical oscillators[37], with the continuous stirred tank reactor being the usual instrument to maintain NESS. In natural (dissipative) supramolecular systems, such as actin filaments or microtubules, NESS conditions are achieved by continuous fuel/waste exchange and compartmentalization by membranes. To mimic the latter approach, we fabricated a continuous flow device consisting of a dialysis cassette (MWCO = 2 kD) clamped in between silicon spacers and quartz windows to form flow chambers flanking the cassette (Fig. 4a,b). Because of the chosen MWCO, **PDI** and **p2-PDI** (either monomeric or polymeric), as well as the two

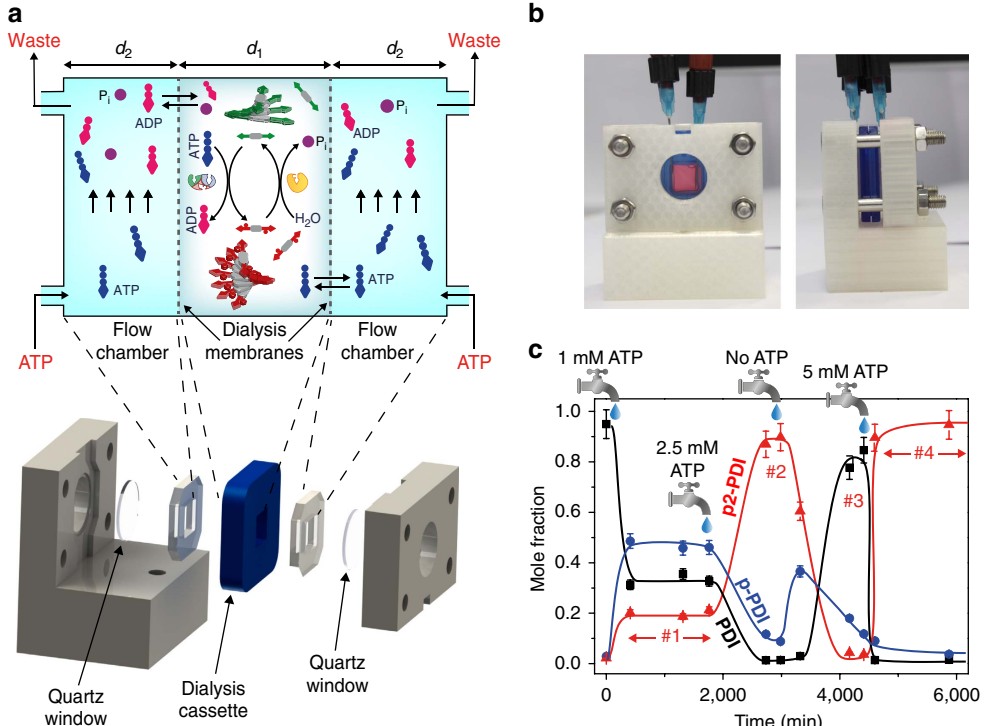

**Figure 4 | Continuous flow device and supramolecular non-equilibrium steady states.** (**a**) Principle of operation and CAD design of the continuous flow device based on a clamped dialysis cassette. The average gap between the membranes $d_1$ is 1.9 mm, the width of the flow chambers $d_2$ 5.1 mm. (**b**) Front and side view of the three-dimensional (3D) printed device, also showing the inlet and outlet needles inserted through the silicon spacers. (**c**) Different NESS (plateau regions, no. $1-4$), characterized by different molar fractions of the three species **PDI**, **p-PDI** and **p2-PDI** (by LC–MS analysis), can be obtained using the continuous flow device, depending on the influx of the fuel ATP. Solid lines are drawn to guide the eye. The taps are placed in correspondence of the time(s) at which we changed the ATP concentration of the solution that is continuously flowed through the lateral flow chambers. Namely, at $t = 50$ min we started to flow ATP 1 mM, at $t = 1,770$ min we switched to ATP 2.5 mM, at $t = 3,170$ min we started to flow buffer without ATP and at $t = 4,440$ min we switched to ATP 5 mM. For all NESS experiments: 0.085 μM PKA and 0.097 μM λPP. The error bars were set to 6% of the corresponding value. The latter is the maximum error observed in the determination of the molar fraction when injecting 3 times in LC–MS solutions of known composition.

enzymes PKA and λPP, are compartmentalized inside the cassette, whereas ATP and the waste (ADP, Pi) can easily pass through the membrane (Fig. 4a). As a result, the fuel can be added to (and waste removed from) the small fluid volume ($\sim 400$ μl) within the two membranes by continuously flushing the flow chambers with fresh ATP solutions.

Using LC–MS we tracked **PDI**, **p-PDI** and **p2-PDI** by sampling the solution in the cassette at different times (Supplementary Discussion). Figure 4c shows that, starting from a solution containing only **PDI** polymers, we could get a first NESS characterized by the presence of all three species **PDI**, **p-PDI** and **p2-PDI** (mole fractions $\sim 0.35$, 0.45 and 0.20, respectively) by constantly flowing 1 mM ATP (Supplementary Fig. 19). Remarkably, we could sustain such a state for $>20$ h (plateau no. 1 in Fig. 4c), that is, for as long as we kept the influx of ATP (and outflux of ADP and Pi) constant. We have to point out that during this time the phosphorylation/dephosphorylation reactions of the PDI derivatives are continuously occurring while consuming fuel. When increasing the ATP concentration to 2.5 mM (second tap in Fig. 4c), the system responded by reaching a different NESS (plateau no. 2 in Fig. 4c) characterized by mostly **p2-PDI** polymers (Supplementary Fig. 20). From this point, we switched to flow only buffer without ATP (third tap at $t = 3,170$ min in Fig. 4c), and as a consequence the system relaxed back to the nonphosphorylated state, that is, mostly **PDI** present (plateau no. 3 in Fig. 4c, and Supplementary Fig. 21). In fact, the absence of ATP causes the phosphorylation to stop, while the

dephosphorylation keeps progressing. The latter demonstrates that a continuous influx of ATP (and removal of waste) is indeed needed to maintain a phosphorylated NESS, which is a continuously dissipating state. Finally, upon flowing 5 mM ATP (fourth tap at $t = 4,440$ min in Fig. 4c) we could once again stabilize a NESS characterized by **p2-PDI** polymers (plateau no. 4 in Fig. 4c and Supplementary Fig. 22), thus demonstrating that our supramolecular system is perfectly reversible when operating in open flow conditions, and it does not suffer from poisoning. In other words, by modulating the influx of ATP, we maintain our supramolecular system in distinct steady states driven by fuel-consuming enzymatic (de)phosphorylation. By the reaction rates obtained from the model, we can estimate that each serine cycles on average 8 times between phosphorylated and dephosphorylated states during the entire experiment shown in Fig. 4c (6,000 min).

We want to stress that our NESS conditions are not just shifting coupled chemical equilibria as described by Le Chatelier's principle (cf., red crosses in Fig. 1a). Instead, we have three species that cannot equilibrate, but can only interconvert in the presence of the enzymes and chemical fuels. More generally, if a fuelled self-assembly process has fast assembly kinetics compared with reaction kinetics, the assemblies are in a local equilibrium state (seemingly obeying Le Chatelier's principle). That is, the fuelled reactions change the average monomer concentration, but the assemblies themselves are determined by the thermodynamics parameters (that is, Gibbs energy, concentration, temperature

and so on). Said more simply, the fuelled reactions change the reservoir of monomers, and the assemblies behave according to the average conditions around them. However, if the self-assembly kinetics are slow with respect to the reaction kinetics, the thermodynamic parameters are of less concern to the system. The assemblies do not have time to establish a local equilibrium, and the system will be dominated by the kinetics of the fuelled reactions instead.

In conclusion, our studies show that it is possible to obtain true non-equilibrium steady states of artificial self-assembling systems depending on the influx of chemical fuel supplied, and outflux of waste products. Unlike in continuous stirred tank reactor systems, where a fraction of the entire system is discarded due to outflow, we developed a membrane reactor that selectively exchanges fuel and waste, but leaves the self-assembling building blocks and costly enzymes in place. Existing systems that currently display transient self-assembly in batch or semibatch conditions can in this way be maintained out of equilibrium indefinitely. It will be very interesting to explore their different steady states depending on how hard the system is driven. In our system, the influx of ATP is the control parameter that determines the nature of the steady state and distance from the thermodynamic equilibrium. In general, if supramolecular systems can be driven in this way, it could give rise to sustained supramolecular oscillations analogously to microtubules[38,39]. Our work opens an avenue towards the development of more life-like self-assembled materials that can show true adaptability and eventually perform functions as complex as in living systems.

## Methods

**General.** The detailed synthesis and characterization of **PDI** and intermediates, the study of phosphorylation by LC–MS and $^{31}$P-nuclear magnetic resonance, the thermodynamic study of supramolecular polymerization, more atomic force microscopy and transmission electron microscopy images, self-sorting experiment, the mathematical model of the enzyme network as well as the mass spectra corresponding to the NESS of Fig. 4c are provided in the Supplementary Methods and Supplementary Discussion.

**Experimental conditions.** The enzymatic reactions were performed, when not differently specified, in 50 mM Tris-HCl buffer (pH 7.5) containing 10 mM MgCl$_2$, 1 mM MnCl$_2$, 2 mM dithiothreitol and 0.1 mM EDTA at 25 °C. We chose to prepare a customized buffer instead of using the standard buffers supplied by the vendor with the enzymes because the latter contains 0.01% of the polyethylene glycol surfactant Brij 35 that is likely to affect the self-assembly of our molecules. The addition of MnCl$_2$ was required for the activity of λPP that is a Mn$^{2+}$-dependent phosphatase. We refer to this buffer as the reaction buffer. Typically, stock 1 mM **PDI** solutions, in freshly prepared reaction buffer, were prepared by accurate weighing and used to obtain the diluted (100–400 μM) **PDI** solutions used in the experiments. The stock solutions were freshly prepared the same day of the experiment.

**Phosphorylation/dephosphorylation.** In a typical phosphorylation experiment, an aliquot of freshly prepared ATP (1–4 μl, 20–50 mM in the reaction buffer) was added to the **PDI** solution (200–400 μl, 200 μM), followed by the addition of PKA (1–3 μl of the supplied 13 μM solution) and gentle shaking. The equivalents of ATP reported in the paper are with respect to **PDI** molecules (that is, 2 eq. of ATP correspond to 1 eq. per phosphorylation site). The dephosphorylation was triggered by the addition of an aliquot of λPP (1–3 μl of the supplied 20 μM solution) to a **p2-PDI** solution, prepared as described above, in the absence of unreacted ATP. The enzymatic reactions were performed in screw cap sample vials equipped with a glass insert (when followed by LC–MS), or in a 1 mm quartz cuvette (when followed spectroscopically by ultraviolet–visible or CD).

**Ultraviolet–visible measurements.** Ultraviolet–visible spectroscopy was used to follow (in time) the changes in aggregation upon phosphorylation (Fig. 2a,b), as well as to study the thermodynamics of supramolecular polymerization by heating/cooling cycles (Supplementary Fig. 10/Supplementary Table 1). To this end, we recorded the absorption spectra of **PDI** (or **p2-PDI**) solutions versus time (or temperature), from which we calculated the ratio of the two maxima at 504 and 540 nm ($A_{504\,nm}$ and $A_{540\,nm}$). Note that the absorbances of the real maxima were considered to calculate $A_{504\,nm}/A_{540\,nm}$ (that is, taking into account the peak shift).

**CD measurements.** CD spectroscopy was used to study the changes in supramolecular chirality upon (de)phosphorylation in both the stepwise (Fig. 2e,f) and batch (Fig. 3b,c) experiments, as well as to study the self-sorting of **PDI** and **p2-PDI** assemblies (Supplementary Fig. 13). For the time-course CD measurements (see, for example, Fig. 2f), the phosphorylation was triggered by the fast addition of small aliquots (∼2 μl) of PKA and ATP solutions to 200 μl of a **PDI** solution contained in a 1 mm quartz cuvette, followed by gentle shaking. Afterwards the CD signal at a fixed wavelength was monitored versus time with time steps of 5 s.

**LC–MS measurements.** Liquid chromatography–mass spectrometry (electrospray ionization–ion trap) was used to both check qualitatively the (de)phosphorylation and to calculate the fraction of the three species **PDI**, **p-PDI** and **p2-PDI** in the batch and steady state experiments reported in Figs 3a and 4c. For the latter, the reconstructed ion chromatogram was obtained for the three species taking account their characteristic ions ($m/z \pm 2$), from which their relative mole fraction were calculated.

**Dynamic light scattering.** A 1 mM **PDI** solution in the reaction buffer was split in two parts, and ATP (4 mM, 4 eq.) was added to both the resulting solutions. After that the **p2-PDI** sample was prepared by adding PKA to one of the two solutions. After complete phosphorylation (run overnight and checked by LC–MS) both samples were measured by DLS. The intensity cross-correlation functions $g^{(2)}(q, \tau) \equiv \langle I_1(q,0)I_2(q,\tau)\rangle/\langle I_1(q,0)I_2(q,0)\rangle$ were measured on a home-built light scattering setup with an ALV7002 digital correlator using a laser diode at $\lambda = 639$ nm at variable scattering vectors $q = \frac{4\pi n}{\lambda}\sin(\frac{\theta}{2})$, where $n$ is the refractive index of the solution. The relaxation dynamics were analysed by the inverse Laplace transformation $g^{(1)}(\tau) = \int_0^\infty H(\Gamma)e^{-\Gamma t}d\Gamma$, where $g^{(1)} = \sqrt{g^{(2)} - 1}$. The translational diffusion coefficient $D$ was calculated with the slope of $\Gamma/q^2$ obtained from experimental $g^{(1)}$. Hydrodynamic radii $R_H$ were then calculated using the Stokes–Einstein relation $R_H = k_B T/(6\pi\eta D)$, where $D = 1/(\tau q^2)$.

**Continuous flow experiments.** The cassette holder in the continuous flow device (white part in Fig. 4a,b) was three-dimensionally printed in polylactic acid. A commercial Slide-A-Lyzer dialysis cassette with MWCO = 2 kD was used as membrane reactor. Two needles connected to microfluidic tubing by Luer lock connectors were inserted through each silicon spacer as the inlet and outlet of two flow chambers flanking the dialysis cassette (Fig. 4a,b). The inlet needle was pushed down to the bottom of the flow chamber while the outlet was kept at the top (visible in Fig. 4b), so to ensure that the chamber was always full and the entire surface of the dialysis membrane was in contact with the ATP solution. The inlet was connected to the ATP (or buffer) reservoir and the outlet going to the waste. Each flow chamber has a volume of ∼800 μl. In a typical experiment 400 μl of a 200 μM **PDI** solution containing PKA (0.085 μM) and λPP (0.10 μM) were injected in the dialysis cassette, and remaining air was carefully removed. After that, the 1 mM ATP solution was flowed through the chambers by using a pressure-driven pump (Elveflow). We started flowing ∼20 ml at a flow rate of ∼4 ml min$^{-1}$, after that we set the flow rate at ∼150 μl min$^{-1}$. At defined times, small samples (∼40 μl) of the reaction solution were taken from the cassette, analysed by LC–MS (5 μl injection) and the remaining part put back in the cassette. At each change of the ATP concentration, the flow chambers were washed with the new solution by flowing at least ∼25 ml at a flow rate of ∼5 ml min$^{-1}$, before setting the flow at ∼150 μl min$^{-1}$.

**Data availability.** All reported data are available from the authors on reasonable request.

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

## Acknowledgements

This work was financially supported by the Marie Skłodowska-Curie Individual Fellowship 'ASSEMZYME' (Grant 658190 to A.So.), Region Alsace, University of Strasbourg Institute for Advanced Study (USIAS), ANR-10-LABX-0026_CSC, FP7 COST action CM1304 and Marie-Curie 'NON-EQ-SA'. We acknowledge the support and the use of resources of the French Infrastructure for Integrated Structural Biology FRISBI ANR-10-INSB-05 and of Instruct, a Landmark ESFRI project.

## Author contributions

A.So. and T.M.H. designed the experiments. A.So. performed the synthesis and experiments, analysed data and performed modelling. J.L.-I. performed synthesis, nuclear magnetic resonance and provided help. T.M.H. performed modelling and three-dimensional printing. A.Sa. performed (analysis of) DLS data. A.So. and T.M.H. wrote the paper. All authors discussed about the results, and commented on the manuscript. T.M.H. conceived the overall project and supervised the research.

## Additional information

**Competing interests:** The authors declare no competing financial interests.

