## [Peer Review File · Nature Communications]

REVIEWERS' COMMENTS:

Reviewer #1 (Remarks to the Author):

The revised manuscript confirms my opinion that these are very nice results, which can be expected to attract significant attention from researchers working in the field of (dissipative) self-assembly. Therefore I am in full support of recommending the manuscript for publication in Nature Communications.

Nevertheless, on this occasion I would like to comment on some of the authors' responses to the reviewers observations. With regard to my own comment on the analogy between this system and the system described by George earlier this year I don't agree with the authors that the systems are fundamentally different (final phrase: 'That is, George performs irreversible reactions on the ATP fuel, but the self-assembly just follows the ATP concentration (i.e., Le Chatelier-type shifting of the binding equilibrium of ATP to building block). In our system, we perform irreversible reactions directly on the building blocks (not Le Chatelier).') However, the system of George can be described using the exact same terminology as used by the authors to describe their own system in Figure 3 of the rebuttal. Phosphorylation of ADP using CPK brings the system to a new energy landscape (ATP+building block), whereas the dephosphorylation using HK drives the system back to the original landscape (ADP+building block). The transition between the energy landscapes is governed by irreversible chemical reactions, just like in the current case. In George's case these processes regulate the effective ATP-concentration which determines the self-assembly process. In the current manuscript, the kinetic processes regulate the concentration of phosphorylated building block, which also determines the outcome of the self-assembly process. In both systems, Le Chatelier is operative as long as the system resides in one energy landscape. In my opinion it's the same principle – which by the way is the same for other (batch-) systems reported recently in the literature. Therefore, I stick to my previous comment that I much appreciate the effort to develop methodology that allows for a continuous operation of this kind of self-assembly processes (which marks an important step forward for the field), but that I don't see a novel chemical insight.

Currently, a strong debate is going on in the community on the definition of out-of-equilibrium systems, which also emerges from the comments by reviewer 3. Without entering into the discussion here, the different views stem from a different perspective on dissipative self-assembly (DSA). According to reviewer 3, in DSA the continuous consumption of fuel is required to push the system out-of-equilibrium (and allow for self-assembly to take place). Leigh's recently published motor works according to this principle and also Credi's system. This is not the case for the system presented here, in which energy is used to maintain the system in a different energy landscape. Under steady-state conditions (rate of phosphorylation = rate of dephosphorylation) the system will reside in a given energy landscape and it's behaviour will be governed by Le Chatelier. Evidently, there is no element of judgment here. Both approaches fall under the category of energy-driven chemical systems, which is generating very exciting opportunities.

Reviewer #2 (Remarks to the Author):

The manuscript is a revised version of the original submission to Nature Chemistry. The topic of non-equilibrium systems is challenging and generating strong interests. The significance, novelty, and the broad appeal of the work are consistent with the standard of Nature Communications, and the authors have addressed the reviewers' concerns properly and intensively. As a result, the publication in Nature Communications is highly recommended.

Reviewer #3 (Remarks to the Author):

The authors have addressed the points I raised to a certain extent. Yes out-of-equilibrium is a tricky issue and some of the papers dealing with it and cited in the paper have, to my opinion, which is shared by others including Astumian, got it wrong.

The authors should fix figure 1a - equilibrium and reaction arrows are not the same!

I will go back to my original note - ATP turnover is not necessary for the self assembly and so no dissipative self-assembly is occurring! In reality self-assembly has nothing to do with ATP. ATP is a reagent that is depleting the monomer (and the reverse process is supplementing it) and driving the equilibrium to the left - Le Chatelier's principle. The self-assembly is governed by K_{eq} and regular thermodynamics. Yes one can play nice games of taking a stable self-assembled structure and destabilizing it through tricks and calling it transient, like many others have done. You can also couple it with another self-assembly but still this doesn't mean that the self-assembled structure is consuming energy and fuel to sustain itself.

I like the waste removing protocol and that will be helpful in other systems and that is why I am fine with paper being published in NatureCommunication. This part is important enough and justifies it.

Time will deal with all the interesting things in the literature pertaining to be out-of-equilibrium! I can also be wrong.

Reviewers Comments:

Reviewer #1 (Remarks to the Author):

Sorrenti et al. report a very nice piece of research on the very timely topic of out-of-equilibrium self-assembly driven by chemical fuels.

Addition of ATP results in the enzyme-catalyzed phosphorylation of peptide-conjugated building blocks, which then spontaneously aggregate into stacks. A second enzyme causes dephosphorylation resulting in disassembly.

Two main novelties are claimed by the authors: the first is the development of a new chiral supramolecular polymer system that operates out-of-equilibrium; the second relates to the development of a membrane reactor that permits the non-equilibrium steady-state to be sustained in time. The experiments are well-carried out and fully support the claims.

Yet, based on the following arguments, I think that the elements of novelty are not sufficiently strong for recommending publication in Nature Chemistry.

1. The system reported here resembles much other systems, in particular the one by Van Esch et al. (refs. 2+3) and George et al. (ref 9.). In particular, the latter one relies also on ATP as a chemical fuel to form supramolecular chiral stacks. Although in ref. 9, George et al did not show out-of-equilibrium self-assembly, this was effectively shown very recently in a follow-up Angewandte paper (10.1002/anie.201610946 – including reversal of supramolecular chirality).

Firstly, the main goal and novelty of our paper consists in demonstrating, for the first time in supramolecular chemistry, the achievement of sustained non-equilibrium steady-states (NESS) of a supramolecular polymer, obtained in a thermodynamically open system (as well emphasized by Reviewer #2). In this regard, we went beyond the current state of the art of fuel-driven transient self-assembly, in which the batch addition of a chemical fuel pushes the system temporarily out-of-equilibrium, for example triggering (dis)assembly (Van Esch, refs. 2+3) or a conformational change (George DOI: 10.1002/anie.201610946), where eventually the system relaxes back to its original state when the fuel is consumed (also referred to as a “batch excursion” in the paper). All the papers cited by the Reviewer are good examples of the latter, as we stressed in the introduction, now also including the very recent Angewandte from George, DOI: 10.1002/anie.201610946. The reason why the George’s paper was not cited in the original version is just because it was first published after our first submission. So, in the revised version of the manuscript we have added this reference together with a short description of the work in the introduction. Note that, we also presented an example of batch excursion in our paper (Fig. 2 main text), but only as a first step towards our final goal of making different NESS.

Secondly, our system is very different from those cited also from a “chemical” point of view. Namely, Van Esch uses a small non-gelating dipeptide diacid that can be activated to the corresponding self-assembling methyl ester by a chemical esterification using methyl iodide (or methyl sulphide) as the “chemical fuel”. The ester tends to hydrolyse, so with time the system slowly relaxes back to the non-assembled state. Repeated addition of the fuel allows obtaining transient assembly/disassembly cycles.

George and co-workers exploit the supramolecular interaction (Zn-complexation) of ATP or ADP with a supramolecular polymer made of naphthalene-dipicolylethylenediamine building blocks, which stabilize either one handedness or the other. Using two enzymes which work in tandem: one producing ATP from ADP and the other hydrolysing ATP to ADP, George and co-workers are able to control in time the ATP/ADP ratio, to obtain “transient helicity”, without significant changes of the size of the assemblies. We also use ATP to drive our system, but we make covalent modifications to the PDI building block upon phosphorylation and dephosphorylation. That is, George performs irreversible reactions on the ATP fuel, but the self-assembly just follows the ATP concentration (i.e., Le Chatelier-type shifting of the binding equilibrium of ATP to building block). In our system, we perform irreversible reactions directly on the building blocks (not Le Chatelier).

2. The main novelty of the paper lies in the construction of a membrane-reactor that permits a continuous addition of ATP and removal of just the waste products – not the building blocks. This permits for the first

time that a supramolecular assembly can be kept in a non-equilibrium state for prolonged periods without sacrificing building blocks, Yet, although very elegantly, no new chemical insights are obtained from this achievement. The conclusion that 'our results demonstrate that it is crucial to remove waste products inevitably formed in chemical fuel-driven assembly processes (abstract)' is not strong enough. It would have made a significant difference in case the authors would have shown that new or different chemical properties could be obtained by sustaining continuously the out-of-equilibrium state (compared to a batch-process).

The dynamics of the systems are completely different, in an open (membrane) vs batch reactor. Only the former allows to keep constant the concentrations of fuel at different levels while continuing removing waste. We extensively demonstrated that only under this conditions, as opposed to the batch ones, the system can be pushed and kept out-of-equilibrium in a sustained way by continuous dissipation, which allowed us to obtain for the first time a NESS of a supramolecular polymer (note that the latter occurs in homogeneous solution, not in a gel, which is a further difference with the cited examples from Van Esch) However, the fact of staying in a NESS does not guarantee that the system exhibits unusual properties. Oscillations or patterns only occur when and if an eventual Hopf bifurcation (if it exists for a given system) is surpassed, that is if the system is pushed far-from equilibrium (we stress that the latter not a synonym of out-of-equilibrium). Moreover, oscillations usually only exist in a very narrow windows of phase space. We believe that not observing (or better not being able to proof) unusual behaviour does not reduce at all the significance of our results. We do of course agree with the reviewer that it would have been very nice to observe oscillations in our system.

Altogether, without any hesitation I would recommend the work for publication in top-ranking chemistry journals such as Angewandte, JACS, or Nature Communications, but in my opinion the conceptual novelty is not strong enough for Nature Chemistry.

In agreement with the suggestion of Reviewer #2, we decided to resubmit to NCOMM.

As a minor issue, I would strongly suggest to make the manuscript more accessible to the reader by removing technical details and numbers from the main text. The model reported in Figure 1 is impressive, but fits much better in the Supporting Information. The complexity of Figure 1 makes it very complicated to grasp the concept.

We have simplified Figure 1 of the main text as suggested by all reviewers. In addition, we have added a new figure in the Supporting Information (i.e., Fig. S1), where we present the complete enzyme network, as recommended by Reviewer #1.

Reviewer #2 (Remarks to the Author):

The manuscript submitted by Hermans and co-workers presented a fuel-driven supramolecular polymer system for the demonstration of sustained non-equilibrium steady-states (NESS). The monomer is based on a perylenediimide scaffold attached with two serine-containing peptide arms. The perylenediimide unit serves as a core for aggregation, while the chosen peptide sequence can be reversibly modified by phosphorylation and dephosphorylation catalyzed by PKA and \square PP, respectively. Moreover, the change of charge state affects the polymerization, especially the stability and CD response. By performing stepwise assay, then batch excursion, and finally the continuous flow analysis with a special membrane, the complexity of the system was built up, and the steady-state (NESS) was regulated by varying the fuel (ATP) concentration.

This is an interesting work of non-equilibrium artificial systems. The systems chemistry of complex mixtures to mimic biology is an intensive and competitive area of research. Different from thermodynamic controlled systems, which have largely been the focus since the birth of supramolecular chemistry, kinetics controlled systems are generating strong interests recently and could lead to new discoveries. Understandably, it is

challenging to devise a completely synthetic non-equilibrium system, and therefore, it is important to take a page from natural systems (i.e., peptide sequence for kinase substrate), which was employed to dictate the supramolecular polymerization. The characterization was carefully conducted, and the theoretical kinetics model was deduced intensively. The approach of continuous flow analysis for adding fuel and removing waste could hold broad appeal.

We thank the reviewer for the nice words on our work and for recognizing that it is novel and timely.

However, there are several issues:

1. At the first glance, Figure 1a is very complicated, especially the enzymatic network, which is actually discussed in the section of batch excursions and Figure 3. It might be better to show this enzymatic network in Figure 3 and simply Figure 1a.

We have revised Figure 1 as suggested, and presented the enzyme network in the new Fig. S1 of Supplementary Information.

2. Line 81: “The introduction of two phosphate groups onto our molecule changes its charge (4+ for PDI vs. zwitterionic 4+/4- for p2-PDI), in turn affecting the structure and stability of their supramolecular polymers”. I don’t completely agree with the statement of the charge state of phosphorylated serine, and whether it is -1 or -2 is pH dependent. At physiological pH the percentage of both states would be significant.

We understand the reviewer concern, and of course we are aware of that. When considering the pKa values for the first and second dissociation of phosphoric acid, i.e. 2.1 and 7.2 respectively, at the used pH of 7.5 (and for p2-PDI concentration of 0.2 mM), it can be easily calculated that all the phosphate groups are at least -1 and up to 67% have a -2 charge. This means that the average charge of each phosphate group is around -1.63. For the entire molecule bearing two arms (with in total 4 arginines and 2 phosphorylation sites), the overall charge changes from 4+ for PDI, i.e. fully cationic, to 4+/3.26- for p2-PDI, i.e. nearly zwitterionic, which has evidently a strong effect on the electrostatic properties of our building-blocks and in turn on the structure and stability of their supramolecular polymers, as found experimentally. We therefore changed the main text to “nearly zwitterionic 4+/3.3-“, instead of simply zwitterionic.

3. Line 102: Both enzymes need to work “at commensurate speeds”. This is an important issue for a non-equilibrium system to work, and such a description is too vague. Based on Figure 2f, the rate of dephosphorylation is much faster than phosphorylation.

Yes, but the experiments in Figure 2 are not performed under the same conditions as compared to the NESS experiments in Figure 4. To clarify this point further, we have added an extra sentence in the caption of figure 4 to point out the conditions: “For all NESS experiments: 0.085 μ M PKA and 0.097 μ M λ PP.”

4. During control experiment and step-wise analysis (between line 106 and line 186), did the authors observe p-PDI? p-PDI is the intermediate between PDI and p2-PDI irrespective of the sequence of reagent addition. The authors didn’t address this issue at all here. In Figure S2, is there any peak for p-PDI? The authors mentioned p-PDI in the caption, but didn’t label it in NMR.

It is true that **p-PDI** is an important intermediate, and we did not emphasize much on this species to keep the main text as simple as possible. The reviewer is right in the labelling of the NMR, but since the ³¹P-NMR shows the phospho-serine residue both **p-PDI** and **p2-PDI** have exactly the same chemical shift. We have updated the labelling in Fig. S2 to reflect this.

Based on data in Figures 4c, S15, and S16 (see more on issue 7), p-PDI is stable enough for isolation and characterization. In order to make a stronger argument, the authors should attempt to prepare and isolate p-PDI and study its aggregation as well as CD properties. It might even be worthwhile to track the kinetics

for independent phosphorylation ($PDI \rightarrow p\text{-}PDI \rightarrow p2\text{-}PDI$) and dephosphorylation ($p2\text{-}PDI \rightarrow p\text{-}PDI \rightarrow PDI$) as well as to establish a kinetics model, analogous to that illustrated for batch excursion.

We do agree with the reviewer that p-PDI would in principle be stable enough to be isolated. However, the kinetics of the different substeps (i.e., $PDI \rightarrow p\text{-}PDI$ and $p\text{-}PDI \rightarrow p2\text{-}PDI$, and the reverse steps) are accurately captured by our kinetic model (cf. Fig. 3a of the main text), so limited additional information is expected from such experiments in terms of kinetics or CD properties.

From my understanding (I could be wrong), for a sequential reaction (such as, $A \rightarrow B \rightarrow C$) it is possible to capture B. That's should be more likely in the current paper, because p-PDI and p2-PDI is the monophosphorylation and diphosphorylation product of PDI, respectively. From the perspective of organic chemistry, p-PDI is a decently stable structure. One natural question to ask is if the component distribution for the phosphorylation is dependent on the equiv. of ATP. The authors always used 2 equiv. of ATP relative to PDI in the main text between line 106 and line 186. How about less or more ATP?

Indeed in the section of the main text (line 106-186) we always use 2 equivalents of ATP, meaning 1 for each arm of the molecule, to have complete phosphorylation of PDI to p2-PDI. This is done to make sure all ATP is consumed before moving on to the (separate) dephosphorylation. In this way we could study an actual "step-wise" phosphorylation/dephosphorylation. If we would use fewer equivalents, this would result in slower and incomplete phosphorylation. When using more (than 2) equivalents of ATP, the phosphorylation is faster, and excess ATP will remain in solution, which in addition decreases the apparent rate of the subsequent dephosphorylation triggered by the addition of the phosphatase enzyme. This is because, as long ATP is present, the two enzymes can work simultaneously and the kinase can phosphorylate back a serine that has been just dephosphorylated by the phosphatase (i.e. two competing processes, instead of two consecutive ones). In the figure below, we show an example of phosphorylation triggered by the addition of PKA and 4 equivalents of ATP (i.e. 2 for each arm of the molecule), followed by dephosphorylation upon addition of λ PP phosphatase, as followed by CD. The phosphorylation is indeed faster when compared with the results shown in Fig 2f of the main text: complete phosphorylation (the plateau) is reached in ~ 150 min versus the ~ 300 min needed when using 2 eq. of ATP. On the other hand, the dephosphorylation clearly show a two phase kinetic behaviour with an initial faster decrease of the CD followed by a slower relaxation to the non-phosphorylated state (because of the competition with the phosphorylation reaction until all ATP is consumed). Overall the complete dephosphorylation to the initial state takes ~ 400 , i.e. it is much slower than when using 2 eq. of ATP (cfr Fig 2f).

The point is that the author should convincingly demonstrate with more experiments that the observation is indeed from NESS, rather than thermodynamic behaviors?

Here, we want to stress that the conversions between all species are absolutely irreversible. That is, phosphorylation only happens by kinase catalysed reaction, and dephosphorylation only by phosphatase catalysed hydrolysis. In other words, there is no chemical equilibrium (Le Chatelier) in between the

different species. Therefore, the relative distribution of species is dictated only by the kinetics of the reactions, and not at all due to their thermodynamic stability. We now realize (also taking into account the discussion with reviewer 3) that this was not sufficiently clear in the manuscript, and we have therefore added the following lines to the main text (see below). Line 108 :“ The conversion between the three species (**PDI**, **p-PDI**, and **p2-PDI**) occurs through irreversible enzyme catalyzed reactions. Thus, the solution composition is only dictated by the relative rates of competing phosphorylation and dephosphorylation, but not by their thermodynamic stability (i.e., the three species are not in chemical equilibrium, cf. red crosses in Fig. 1a). Moreover, to have sustained NESS it is crucial to have fast and completely reversible cycles of (de)activation. Otherwise, the reaction cycle **PDI**→**p-PDI**→**p2-PDI**→**p-PDI**→**PDI** would stop after several runs.” In addition, we have emphasized this aspect schematically in the new version of Figure 1 and further explained it in its caption.

5. The author’s description of batch excursion appears to be inconsistent. In Figures 1 and 3, it stated “the addition of ATP, PKA and PP” to PDI. In line 192, it stated “upon addition of a shot of ATP (c = 1 mM, 10 eq.) to a solution containing PDI (c = 100 μM), PKA (c = 0.13 μM) and λPP (c = 0.3 μM)”.

This is indeed a good point of the reviewer. We have now unified this notation in the entire manuscript. Specifically we changed the main text / captions using the formula “...the addition of ATP (.....) to a μM **PDI** solution containing PKA (.....) and λPP (.....)”, which describes better the actual experimental methodology followed.

6. The author’s claim that “in addition, upon ATP-fueled phosphorylation the supramolecular chirality of the polymer is reversed” in the abstract (line 7) is mainly based on the changes in CD spectra (Figure 2e) and the comparison with literature (27-29, line 164). There is the same issue with Figure S9 and S10 (self-sorting). A mixture of PDI/p2-PDI was employed for Figure S9, and it is difficult to interpret Figure S10 without the information about p-PDI.

It is true that we did not isolate p-PDI and study its CD properties, but as mentioned above, we expect limited new insight from such studies. We do want to stress that CD in our work is not used quantitatively to describe species kinetics, but just to follow qualitatively the process. On the other hand, we have mainly used LC-MS to identify the composition of the system at all times. That is, all three species can accurately be quantified by LC-MS. In addition, the fact that a smooth kinetic curve is observed during the step-wise phosphorylation (Fig. 2f, and CD vs time curve presented above) points out that the CD properties of p-PDI are intermediate to those of PDI and p2-PDI. A full study of p-PDI in this context is beyond the scope of this first manuscript on our (entirely new) system, and we believe the main emphasis of the work (i.e., NESS) is demonstrated well using LC-MS data. Lastly, in the self-sorting experiment of Fig. S9 (that is now Fig. S10 in the revised paper), different mixtures PDI/p2-PDI were prepared starting from the pure samples. We stress again that in the absence of ATP and λPP (needed respectively for phosphorylation and dephosphorylation) the two compounds can neither interconvert among them nor form p-PDI, which was the case for that experiment.

7. For LC-MS data in Figures 15-18, the results are from a mixture. That could be misleading since p-PDI could be created due to the fragmentation of p2-PDI under ESI-MS condition. It is better to include both LC trace and the MS from the individual peak within the LC trace. I guess that it is helpful to have a standard sample of p-PDI for the comparison of LC retention time.

At first glance we agree that it might be difficult to accurately identify p-PDI as a separate species vs. fragmentation in the LC-MS, but we have 100’s of LC-MS on pure p2-PDI, and have never observed m/z signals characteristic of p-PDI. See for instance Fig S1b and S3a (now Fig. S2b and S4a), which show MS spectra for the pure compounds p2-PDI that have no traces of p-PDI. Also Fig S19 shows that during a NESS with high ATP concentration (5 mM) we only observed p2-PDI without any fragmentation into p-PDI (which

does not occur under the used conditions). In other words, p-PDI showing up due to fragmentation is not occurring under our conditions.

In summary, this reviewer believes that the significance, novelty, and the broad readership of this manuscript are consistent with the standard of Nature Chemistry, but several issues exist. As a result, the decision about possible publication on Nature Chemistry can only be made after the authors address those concerns.

We thank reviewer #2 for his/her constructive suggestions, and hope we have now clarified the manuscript sufficiently. We do appreciate the trust in the quality of the work for NCHEM, but in agreement with the editor we have decided to resubmit to NCOMM.

Reviewer #3 (Remarks to the Author):

This is a nice paper from Herman's et al showing how a phosphorylation/dephosphorylation of an NDI containing polymer can be used in changing its helical supramolecular self-assembly. The authors show using different spectroscopies and analysis that going from one structure to another is associated with a change in chirality and size. Moreover, there is indication that there is self-selection, i.e., the phosphorylated systems self-assembles with itself and no assembly happens with the non-phosphorylated system. In order to switch the system multiple times the authors built a continuous flow system with appropriate filters to make sure that the catalyst (enzyme responsible for phosphorylation/dephosphorylation) is not poisoned by the waste product. The results, especially the switching of the chirality, are very interesting and the conclusions are supported, to a certain extent, by the data in the paper. Having said, it is in my opinion that the paper will be better suited in another high impact journal but not Nature Chemistry for the reasons given below.

We thank reviewer #3 for the nice comments, and hope that our revision (see below) will clarify his/her concerns.

The main point of this paper is that the self-assembly is out of equilibrium and fueled by ATP, and I have reservations about these two claims. First of all the phosphorylated self-assembled structure is stable by itself, as the authors stress on page 4. Hence, there is no need for a fuel to sustain it (the engine is one even without gas)! For a system to be an out of equilibrium self-assembly, a turnover of fuel is required to sustain the self-assembled structured, which otherwise would break down.

We demonstrate in Fig 4 that the out-of-equilibrium steady state NESS can indeed only be maintained by fuel consumption. If ATP supply is stopped, then the system relaxes to its equilibrium state (that it is also a demonstration that it is out-of-equilibrium). The fuel ATP is indeed needed (and at the sufficient level) to maintain the phosphorylated state in the presence of the kinase/phosphatase system, and at the same time it is necessary to remove waste (sustained NESS). It is true that in absence of phosphatase p2-PDI is a stable chemical species, but our description is not about structural stability in isolation, but about the stability of the system as a whole (i.e., including all species and enzymes). We have also stressed this point further in the revised version of the manuscript.

This is not the case here. To argue that this system is out of equilibrium, is akin to arguing that every reaction is out of equilibrium until it reaches equilibrium. What the authors are doing here is simple playing around with Le Chatelier's principle and shifting a reaction from one side of an equilibrium to another by adding an external reactant, which is ATP in this case. If this is out-of-equilibrium assembly/reaction then it is a very very trivial one, and a plethora of similar ones can be easily devised. For example, the author can take any of beautiful metal (let's say zinc(II)) containing self-assembled systems developed in Strasbourg, and add a large amount of cryptofix to them. If the binding constants are well thought of then the dis-assembly will be slow enough to allow someone to start spiking ("fueling") the system with zinc(II) and

shifting the equilibrium towards assembled structure for a while. Again playing with Le Chatelier's principle to shift an equilibrium but no self-assembly of structure that is out of equilibrium that requires fuel to sustain this structure. Let's take another reaction: acid catalysed dehydration – any one you like – while doing the reaction you make sure to remove the water from the system otherwise the reaction will go back to hydrated state. Let's say that now while the reaction is happening you dump a quantity of water to the solution by mistake. This will of course shift the reaction/equilibrium towards the hydrated form. Is this now an out of equilibrium reaction?

To clarify better this point we have written a short paragraph (see below) to demonstrate that Le Chatelier principle cannot be applied to our NESS, the latter obtained working in a thermodynamic open system.

Le Chatelier versus non-equilibrium steady states (NESS)

The definition of Le Chatelier's principle is¹: "A system at equilibrium, when subjected to a disturbance, responds in a way that tends to minimize the effect of the disturbance."

The latter principle is of great use in chemistry, and has made it possible to rationalize systems that are at equilibrium, or disturbed from their equilibrium state. Referee #3 draws analogies of our NESS with acid catalyzed dehydration ("any one you like"). We try to illustrate the differences between Le Chatelier and NESS by using the dehydration of an alcohol as an example (Fig. 1).

Fig. 1 | Acid catalyzed dehydration of cyclohexanol: thermodynamic considerations. $Q = ([\text{cyclohexene}][\text{water}]/[\text{cyclohexanol}])$ at a certain time t , and $K_{eq} = ([\text{cyclohexene}][\text{water}]/[\text{cyclohexanol}])$ at thermodynamic equilibrium.

The dehydration starts with cyclohexanol (#1, Fig. 1) and proceeds spontaneously (i.e., exergonic, $\Delta_r G < 0$) until it reaches the thermodynamic equilibrium (#2, Fig. 1), where there is no more driving force to continue (i.e., $\Delta_r G = 0$). Importantly, Le Chatelier can be used to explain the dehydration because there is microscopic reversibility. That is, the forward and backward reactions follow the exact reverse pathway.²

Fig. 2 | Gedankenexperiment: sudden addition of H₂O to dehydration of alcohol under Dean–Stark conditions. (see definition of Q and K_{eq} in caption Fig. 1)

As referee #3 suggests, the dehydration reaction can be done while removing water (e.g., in a Dean–Stark setup), which causes the equilibrium to shift more to the product side (i.e., more cyclohexene), in accordance to Le Chatelier’s principle. Reviewer #3 suggest the following experiment: “Let’s say that now while the reaction is happening you dump a quantity of water to the solution by mistake. This will of course shift the reaction/equilibrium towards the hydrated form. Is this now an out of equilibrium reaction?”

Looking at Fig. 2, we can see that indeed a transient shift of the composition will result due to a sudden addition of water (green dashed arrow), moving more to the reactant side (i.e., Le Chatelier) up to a certain point (* in Fig. 2). Thereafter, the Dean–Stark removal of water continues (red arrow, Fig. 2), first crossing the thermodynamic equilibrium point (#2, Fig. 2), and continuing to the product side (black arrow, Fig. 2). Note that this dehydration gedankenexperiment works because there is microscopic reversibility of the reaction (see mechanism of dehydration on page 592 of ref. 2).

Fig. 3 | Energy landscape of PDI (green) and p2-PDI (red) system. Double arrows show microscopic reversibility (i.e., equilibria); curved arrows show irreversible reactions.

The latter gedankenexperiment does indeed seem similar to our batch excursion experiment in Fig. 3 of the main text. That is, a transient change in composition is observed. However, our system is fundamentally different, due to the lack of microscopic reversibility in the chemically fuelled steps, i.e.: i) phosphorylation of PDI \rightarrow p2-PDI using ATP catalyzed by PKA, and ii) dephosphorylation of p2-PDI \rightarrow PDI catalyzed by lambdaPP. Both the latter reactions are irreversible!

What in fact is happening in our system, is that there are two distinct molecules (PDI and p2-PDI), with their respective energy landscapes (green and red parabola, respectively, Fig. 3) that are interconverted by irreversible reactions (curved arrows, Fig. 3). Within one energy landscape (e.g., only on the red curve, Fig. 3) Le Chatelier indeed holds for the self-assembly of PDI into the PDI supramolecular polymer, and likewise for p2-PDI monomers forming their p2-PDI polymers.

However, Le Chatelier does not hold for our system as a whole! There is no reversible chemical reaction that connects PDI and p2-PDI together, which is necessary

to allow equilibration, and thus prevents the use of Le Chatelier.

Instead, our system is an open system in a non-equilibrium steady state (NESS), whose composition depends not on the Gibbs energies of the molecules (or assemblies), but instead on how hard the system is driven. The latter referring to the phosphorylation and dephosphorylation rates. In other words, the rates of the enzymatic reactions dominate the system, and not the thermodynamic stabilities of the involved species. See also an excellent insight by Dean Astumian recently, who clearly makes this point.³

As stated by I. Prigogine and D. Kondepudi in a classical textbook of thermodynamic (ref.4, page 243): “*Le Chatelier’s principle only describes the response of a system in thermodynamic equilibrium; it says nothing about the response of a system that is maintained in a nonequilibrium state.*”

Moreover, according to Reviewer #3: “*...the phosphorylated self-assembled structure is stable by itself, as the authors stress on page 4. Hence, there is no need for a fuel to sustain it (the engine is one even without gas)! For a system to be an out of equilibrium self-assembly, a turnover of fuel is required to sustain the self-assembled structured, which otherwise would break down. This is not the case here.*”

It is true that in an isolated system (without enzymes) both PDI and p2-PDI polymers are stable at equilibrium with their respective monomers. However, in our NESS conditions, the presence of lamdaPP causes p2-PDI polymers to be completely disassembled if the influx of ATP is stopped. This was demonstrated in Fig. 4c of the main text, where at $t = 3000$ min. the fraction of p2-PDI drops immediately. That is, p2-PDI is not stable in the system as a whole, so we do need *gas* (in the metaphor of Reviewer #3) to maintain p2-PDI. We can sustain different compositions in steady states, depending on the influx of fuel (ATP) and removal of waste (ADP, Pi).

In short, our approach is the first reported NESS in any artificial supramolecular system relying on chemical fuels. We agree that it is conceptually difficult to understand at first sight, which is in part because we are still in the near-equilibrium regime.⁵ This means, that we do not (yet!) observe so-called emergent properties (or “*different chemical properties*” as referee 1 calls it) like oscillations or reaction–diffusion phenomena (e.g., patterns, waves, etc.), because we have not reached a bifurcation point. This is common knowledge in non-linear chemical dynamics (e.g., when studying chemical oscillators)⁶, but less familiar in supramolecular chemistry. The dehydration example can never lead to emergent properties, due to microscopic reversibility.

However, we strongly believe our approach is the way forward in the field of dissipative non-equilibrium self-assembly, where so far only transiently dissipative systems have been reported.^{7–10} Even a prominent researcher, such as Ben Feringa, uses the outlook of continuously operating systems to point to the future (quote: “*...towards the ultimate goal of the design of an autonomous continuously operating chemically fuelled catalytic molecular motor.*”).¹¹

Under (continuously dissipative) NESS conditions, the currently reported transient systems can in principle cross into the (real) far-from-equilibrium regime, where emergence can occur. We hope that you are convinced of the novelty of our approach, and hope we can bring this to the broad readership of NCHEM.

- 1 P. W. Atkins and J. De Paula, *Atkins’ Physical chemistry*, W.H. Freeman, New York, 2006.
- 2 E. V. Anslyn and D. A. Dougherty, *Modern Physical Organic Chemistry*, University Science Books, 2006.
- 3 R. D. Astumian, *Faraday Discuss.*, 2017, **195**, 583–597.
- 4 D. Kondepudi and I. Prigogine, *Modern Thermodynamics: From Heat Engines to Dissipative Structures*, John Wiley & Sons, 2014.
- 5 Nicolis, G., Prigogine, I. & others. *Self-organization in nonequilibrium systems*. **191977**, (Wiley, New York, 1977).
- 6 I. R. Epstein and J. A. Pojman, *An introduction to nonlinear chemical dynamics: oscillations, waves, patterns, and chaos*, Oxford University Press, New York, 1998.
- 7 S. Maiti, I. Fortunati, C. Ferrante, P. Scrimin and L. J. Prins, *Nat. Chem.*, 2016, **8**, 725–731.
- 8 J. Boekhoven, W. E. Hendriksen, G. J. M. Koper, R. Eelkema and J. H. van Esch, *Science*, 2015, **349**, 1075–1079.
- 9 C. S. Wood, C. Browne, D. M. Wood and J. R. Nitschke, *ACS Cent. Sci.*, 2015, **1**, 504–509.
- 10 S. Dhiman, A. Jain and S. J. George, *Angew. Chem.*, 2017, **129**, 1349–1353.

Now to some other comments:

1) *Figure 1a is too complicated and difficult to follow.*

Yes, we agree with Reviewer #3 on the complexity of Figure 1 and have greatly simplified it.

2) *Figure 2a – why is the absorption starting at -.05 and below? This is a bit strange. But more importantly, it is not clear if the self-assembled structures have absorption at 639 nm or not. This is very important because if they do then the DLS data and following discussion might not be accurate as the self-assembled system will be also absorbing the light thus skewing all the obtained data, and hence, discussion of change in size when going from one structure to another.*

We agree that negative absorption is indeed impossible. We believe the error came from improper baseline correction, and have redone the experiment reported in Fig. 2a and corresponding Fig. 2b, which has now been updated in the revised version of the manuscript. It is now more evident that there is no absorption at the wavelength used for the DLS data (639 nm), and therefore we are confident that the observed change in size (PDI vs. p2-PDI) is real.

3) *Did the authors try to simulate the CD spectrum to make sure that there is indeed an inversion of helicity? This system is different enough from others in the literature (charges that can effect dipoles) that a double check is warranted. On another note Reference 27 does not talk about the CD spectrum of the PDI assembly and a better reference will be J. AM. CHEM.SOC. 2004, 126, 10611-10618*

We did not attempt to simulate the CD spectrum, but it is known from literature that the observed bisignate CD spectra for assembled PDI derivatives are indicative of excitonic coupling between rotationally displaced PDI chromophores in helical stacks, featuring a preferential handedness. An example of a PDI derivative bearing charged alanine-spermine arms, where the bisignate band has been attributed to the formation of chiral stacks, has been reported by Wurthner and co-workers *Chem. Sci.* 2012, **3**, 3393.

We understand the concerns of Reviewer #3, but the charged moieties in our molecules are far away from the PDI core, and we therefore expect no influence. In addition, several studies on DNA-templated chromophores showing excitonic CD have shown helix inversion at very different pH (and thus very different state of charge). See for example the work by Albert Schenning and co-workers (<http://dx.doi.org/10.1002/anie.200903507>). Moreover, in the recent work by George and co-workers on NDI derivative (now also included in the revised manuscript) the inversion of the CD spectrum has been interpreted in terms of transient helix inversion for the NDI stacks upon complexation with ADP or ATP (i.e. different state of charge), <http://dx.doi.org/10.1002/ange.201610946>.

We do agree that the suggested reference *JACS* 2004, 126, 10611-10618 is more appropriate compared to the old ref. 27 in the main text, the latter was misintroduced by mistake in place of those from Wurthner cited above. We have replaced the wrong reference with correct one.

4) *Figure 3a. Calling the match between the experimental data and the ODE model “good” (page 6 lone 214) is a bit of wishful thinking, which puts the whole discussion of this section and conclusions drawn from it into question.*

We agree with the reviewer calling a fit of a model “good” is not very quantitative. What we meant to say is considering a unique solution of 55 ODE's (using a global optimization routine starting from many different random initial values), which shows the essential features and fits onto the experimental data is very

“good”. We do want to stress that the finding of the NESS states (i.e., the main claim of the paper) is not based on the model, but only on experimental observations of species concentrations by LC-MS.

5) *Figure 3c – page 7, 238. $T = 2160$ min is mentioned and it is not clear what should be seen at this point in the figure!?*

What was meant here is that after the 2nd phosphorylation event, which did not result in dephosphorylation due to poisoning (cf. Fig. 3c) we could achieve dephosphorylation by doing dialysis using the same exact sample taken at time $t = 2160$ min. In other words, nothing interesting can be seen in figure 3c at time $t = 2160$ min, but rather that is when the sample was taken out for dialysis. To clarify, we have moved the icon for p2-PDI in Fig. 3c a bit to earlier time to avoid confusion.

This completes our reply to the three reviewers. We have made all modifications to the main text using “track changes” in MS Word, so it is easy for the reviewers to see what has been changed. To be more in line with recent literature we have changed all instances of “batch excursion” to “transient states” or “transient experiments” (van Esch nomenclature).

We are open to any further suggestions by the reviewers, and thank them for their time and effort to thoroughly review our work.

With kind regards,
Thomas Hermans